# Membrane Localization of HspA1A, a Stress Inducible 70-kDa Heat-Shock Protein, Depends on Its Interaction with Intracellular Phosphatidylserine

**DOI:** 10.3390/biom9040152

**Published:** 2019-04-17

**Authors:** Andrei D. Bilog, Larissa Smulders, Ryan Oliverio, Cedra Labanieh, Julianne Zapanta, Robert V. Stahelin, Nikolas Nikolaidis

**Affiliations:** 1Department of Biological Science, Center for Applied Biotechnology Studies, and Center for Computational and Applied Mathematics, College of Natural Sciences and Mathematics, California State University Fullerton, Fullerton, CA 92831, USA; abilog@fullerton.edu (A.B); l.smulders@csu.fullerton.edu (L.S.); rolive1011@csu.fullerton.edu (R.O.); cedralabanieh@csu.fullerton.edu (C.L.); jzap@csu.fullerton.edu (J.Z.); 2Department of Medicinal Chemistry and Molecular Pharmacology and the Purdue University Cancer Center, Purdue University, West Lafayette, IN 47907, USA; rstaheli@purdue.edu

**Keywords:** heat-shock proteins, lipid–protein interactions, phosphatidylserine, plasma membrane

## Abstract

HspA1A is a cytosolic molecular chaperone essential for cellular homeostasis. HspA1A also localizes at the plasma membrane (PM) of tumor and stressed cells. However, it is currently unknown how this cytosolic protein translocates to the PM. Taking into account that HspA1A interacts with lipids, including phosphatidylserine (PS), and that lipids recruit proteins to the PM, we hypothesized that the interaction of HspA1A with PS allows the chaperone to localize at the PM. To test this hypothesis, we subjected cells to mild heat-shock and the PM-localized HspA1A was quantified using confocal microscopy and cell surface biotinylation. These experiments revealed that HspA1A’s membrane localization increased during recovery from non-apoptotic heat-shock. Next, we selectively reduced PS targets by overexpressing the C2 domain of lactadherin (Lact-C2), a known PS-biosensor, and determined that HspA1A’s membrane localization was greatly reduced. In contrast, the reduction of PI(4,5)P_2_ availability by overexpression of the PLCδ-PH biosensor had minimal effects on HspA1A’s PM-localization. Implementation of a fluorescent PS analog, TopFluor-PS, established that PS co-localizes with HspA1A. Collectively, these results reveal that HspA1A’s PM localization and anchorage depend on its selective interaction with intracellular PS. This discovery institutes PS as a new and dynamic partner in the cellular stress response.

## 1. Introduction

HspA1A is the major stress inducible seventy-kilodalton heat-shock protein (Hsp70) in humans and mice, and is essential for cellular homeostasis and survival under both stress and pathological conditions [1]. In addition to its critical functions as a cytosolic molecular chaperone, HspA1A is also found at the plasma membrane (PM) of stressed and cancer cells [2,3]. Membrane-bound HspA1A plays important biological roles, as it activates the immune system, mediates clathrin-independent endocytosis, facilitates viral entry, and regulates tumor cell survival [2,4,5,6].

HspA1A does not contain known transmembrane or secretory signals and therefore its presence at the PM and the extracellular medium has been puzzling. Towards elucidating this conundrum, several reports revealed that HspA1A does not follow the classical PM-localization and secretion pathways [3,7,8,9]. These reports revealed that HspA1A’s PM-localization could be explained by its interaction with lipid rafts while its secretion is accomplished via exosomes [3,7,8,9,10]. The interaction of HspA1A with lipid rafts, which suggests a lipid-mediated anchoring mechanism, is further supported by ex vivo and in vitro studies revealing the interaction of HspA1A with specific membrane lipids [5,7,11,12,13,14,15,16,17,18,19]. These lipids include among others globotriaosylceramide (Gb3), a lipid found primarily in lipid-rafts [7], and phosphatidylserine (PS), a lipid found primarily in the inner leaflet of the PM during normal cell growth [5,12,13,15,16,18,20].

In the case of Gb3, it was established that its interaction with HspA1A enables the protein to localize at the surface of tumor cells as both the lipid’s and the protein’s concentrations increase [7]. Regarding the interaction of HspA1A with PS, several in vitro studies have demonstrated the binding properties of recombinant HspA1A and PS-composed liposomes [12,13,15,16,18,19,20,21]. In addition to these in vitro studies, the interaction between HspA1A and PS has also been shown in human colon and pancreas carcinoma cells, pre-apoptotic PC12 cells, and HepG2 cells. However, in all these studies, the interaction was studied after PS had flipped to the extracellular leaflet of the PM [5,7,16,17]. Therefore, whether HspA1A binds to intracellularly localized PS and whether this binding is related to the protein’s PM localization in non-apoptotic cells both remain unknown.

Based on the abovementioned studies and the well-established notion that lipids recruit specific proteins to the PM [22], we hypothesized that HspA1A selectively interacts with PS, and this interaction allows the chaperone to localize and anchor at the PM. To test this hypothesis, we determined whether a known PS-biosensor [23], the C2 domain of β-lactadherin (Lact-C2), inhibits the translocation of HspA1A to the PM after mild heat-shock using a known PI(4,5)P_2_ biosensor as a control. Further, neutralization of the anionic charge of the plasma membrane inner leaflet and co-localization of HspA1A with a fluorescent PS molecule lend further credence to the central hypothesis. Taken together, the results strongly suggest that PS is the main determinant of HspA1A plasma membrane binding following heat-shock stress in a number of cell lines.

## 2. Materials and Methods

### 2.1. Generation of Recombinant DNA Clones

The cDNA clone containing the *hspA1A* gene sequence, accession number BC054782, was used to generate the recombinant clones used in this study. HspA1A was tagged with either green fluorescence protein (GFP) or red fluorescence protein (RFP). For expression in mammalian cells, the gene was subcloned into the pEGFP-C2 or mRFP-C1 vectors using directional cloning after PCR amplification and restriction digest (see [18] for the complete protocol). The primer sequences used, as well as the restriction enzymes (XhoI and BamHI; underlined nucleotides), which were incorporated for directional cloning, were CATCTCGAGCATGGCCAA-GAACACGGCG and CCGGGATCCATCCACCTCCTCGATGGT [24,25]. Lact-C2-GFP, Lact-C2-mCherry, and R-pre-GFP were a gift from Sergio Grinstein (Addgene plasmid # 22852 and # 17274, respectively) [23]. GFP-C1-PLCδ-PH was a gift from Tobias Meyer (Addgene plasmid # 21179) [26].

### 2.2. Cell Culture, Transfection, and Treatments

Human embryonic kidney cells (HEK293; ATCC^®^ CRL-1573^™^), Henrietta Lacks’ ‘Immortal’ cells (HeLa; ATCC^®^ CCL-2^™^), and liver hepatocellular carcinoma cells (HepG2; ATCC^®^ HB-8065^™^) were purchased from the American Type Culture Collection (ATCC, December 2016). These cell lines were maintained in a humidified 5% CO_2_ atmosphere at 37 °C in complete medium consisting of Dulbecco’s Modified Eagle Medium (DMEM; HEK) or Minimum Essential Media (MEM; HeLa and HepG2) supplemented with 10% fetal bovine serum, 2 mM L-glutamine, and penicillin-streptomycin (and 0.1 mM non-essential amino acids and sodium pyruvate for HeLa and HepG2). A day before transfection, cells were split into 24-well plates at 2.0 × 10^4^ cells/well containing poly-D-lysine-treated coverslips or in 60 mm cell culture plates at 5.0 × 10^5^ cells/plate. After 18 h, cells were transiently transfected with the appropriate construct using the PolyJet In Vitro PNA Transfection Reagent (SignaGen, Rockville, MD, USA) as per the manufacturer’s instructions. Transfection was allowed to continue for 18 h, and then transfection media was removed and replaced with fresh complete media. At that time, cells remained at 37 °C or were placed in a water-bath at 42 °C for 60 min followed by recovery at 37 °C for 8 h. For the transfections on coverslips, 0.5 μg DNA was used for HspA1A or EGFP and 0.1 μg DNA for LactC2 or GFP-C1-PLCdelta-PH. For the 60 mm plates, 2.5 μg DNA was used for HspA1A-GFP and RFP and 0.5 μg for LactC2-mCherry. For the 75 cm^2^ flasks, 9.6 μg DNA was used for HspA1A-RFP and 2.4 μg for GFP-C1-PLCdelta-PH.

To mark the cell edges, the PM was stained using 500 ng/mL wheat germ agglutinin lectin Alexa Fluor^®^ 555 conjugate (WGA-AF555; Life Sciences, Waltham, MA, USA). The coverslips were then mounted on slides using DAPI Fluoromount-G^®^ (SouthernBiotech Birmingham, AL, USA).

To neutralize the PM charge, we used D-erythro-Sphingosine (Avanti Polar Lipids Inc.; Alabaster, AL, USA). The lipid was dried under N_2_ and resuspended in ethanol. Cells were treated with 75 μM sphingosine or ethanol vehicle (1:250 vol/vol) for 12 h at 37 °C.

To observe the intracellular localization of PS and test its co-localization with HspA1A, we used 16:0-(11-TopFluor)PS (Avanti Polar Lipids Inc.). Specifically, to add TopFluor-PS to cells, the required amount of stock lipid in chloroform was dried under N_2_, re-suspended in methanol, and added to phosphate-buffered saline (PBS) containing 3 mg/mL fatty acid free bovine serum albumin (BSA). A final concentration of 1 μM lipid was incubated with the cells for 30 min at 37 °C followed by three washes with PBS and mounting on slides. These experiments were performed only at 37 °C, because TopFluor-PS is a non-metabolized PS analog, which cannot be modified by heat-shock.

To determine apoptosis after heat-shock and other treatments, 8 × 10^5^ cells/mL were dissolved in ice-cold PBS and 25 µL of the cell suspension was placed into a white, clear-bottom 96-well plate (Corning, Corning, NY, USA) in duplicate. Subsequently, 25 µL of Caspase-3/7 Glo (Promega, Madison, WI, USA) reagent was added to the cell suspension and incubated for 1 h at room temperature in the dark. After the incubation, luminescence was measured using the GloMax^®^ 96 Microplate Luminometer (Promega). Staurosporine (STS; Enzo Life Sciences, Farmingdale, NY, USA) was used at 1 μM for 4 h at 37 °C as a positive control for cellular apoptosis.

### 2.3. Confocal Microscopy and Image Analysis

Cells were visualized using a Leica DM IRE2 inverted scanning confocal microscope equipped with a 63× 1.4 oil objective. From the Leica software, raw image files are split and collected as three channels: channel 0 (green (excitation 488 nm; emission 500 nm (bandwidth: 65)), channel 1 (red (excitation 532 nm; emission 590 nm (bandwidth: 75)); and channel 2 (blue (excitation 405 nm; emission 430 nm (bandwidth: 60)).

Image analyses were performed using several cells (number of cells is provided in the figure legends) from three independent experiments. Images were analyzed in ImageJ [27] by the corrected total cell fluorescence (CTCF) method [24,28]. As an example, we provide below the measurement of the CTCF values from images containing HspA1A-GFP (channel 0; green), WGA-AF555 (channel 1; red), and DAPI (channel 2; blue). Images files were loaded in ImageJ and a region of interest (ROI) was determined using the free-hand ROI tool and tracing the plasma membrane in channel 1 (red). Under our experimental conditions, the width of the PM as marked by the WGA-AF555 stain was 5.9 (SD = 1.7) pixels. Therefore, we decided to define the PM by the six outer pixels of each cell as performed by Johnson et al. [29]. This ROI was duplicated onto the image derived from channel 0 (green) of the same cell. The area, mean fluorescent intensity, standard deviation, min, max, and integrated density of the ROI were measured, using the built-in measuring program (Analyze -> Measure), and data were collected. The same process was then repeated to measure the cytosol, by using the channel 0 (green) image to determine the entire cytosol of the cell. Lastly, the background measurement was taken by averaging the fluorescent intensity of the GFP-HspA1A image background in three 20-pixel radius circles, using the provided ROI tool in ImageJ. CTCF was then calculated for the plasma membrane and cytosol by the following formula:

CTCF = Integrated Density − (Area of Region of Interest * Fluorescence of background reading)

A ratio between the CTCF measurements of the PM and the rest of the cell (PM + Cytosol) was then calculated [29].

Because the cell slices used are not completely flat and the method uses the outer pixels of each cell, a value of 0% PM localization cannot be achieved [29]. Therefore, to determine the basal CTCF value, we used a cytosolic protein (EGFP), which, when transfected into HeLa cells, results in a CTCF value of 0.15 (SD = 0.032). In comparison, the CTCF value of HspA1A is 0.21 (SD = 0.025), which is significantly different from the EGFP value (*P* < 0.0001). For comparison, we also calculated the CTCF ratios of the PM marker WGA-AF555, as well as the lipid biosensors Lact-C2 and PLCδ-PH.

Co-localization analyses were performed from the total image Z-projection using the Intensity Correlation Analysis (ICC) plugin in ImageJ [27] for 30 cells from three independent experiments (error bars = SD). The ICC plugin generates Mander’s coefficients [30], which were used to assess co-localization. Furthermore, the plugin performs ICC analysis as described by Li et al. [31]. Images from the latter analysis are included in the figures presented in this report as examples of co-localization instead of merged channel images (see Appendix A as an example). These pseudocolored images are generated by merging two images produced by two different channels (e.g., channel 0 (green) and channel 1 (red)), where each pixel is equal to the Product of the Differences from the Mean (PDM) value at that location. The PDM values are calculated using the following formula:

PDM = (channel 1 (red) intensity − mean channel 1 (red) intensity) × (channel 0 (green) intensity − mean channel 0 (green) intensity)

For clarity, we are displaying only the +ve PDM values resulting from both pixels being above the mean (i.e., red intensity-mean red intensity and green intensity-mean green intensity are both positive).

### 2.4. Cell Surface Biotinylation and Total PM Protein Isolation

To investigate further the PM localization of HspA1A, we performed cell surface biotinylation. In these experiments, we transfected the cells with HspA1A-GFP and RFP-C1 (empty) vector, as well as HspA1A-GFP and Lact-C2-mCherry. For these experiments, immediately following the heat-shock, HEK293 cells were trypsinized, washed three times in PBS (pH 8), and incubated in freshly prepared PBS (pH 8) containing 10 mM CaCl_2_ and 1 mg/mL Sulpho-NHS-LC-biotin (Thermo Scientific Pierce, Waltham, MA, USA) for 30 min at room temperature with constant agitation. The reactions were performed using trypsinized and washed cells to decrease the chances of biotinylating exogenous HspA1A and at room temperature to decrease endocytosis of the used cell-impermeable biotin. The reactions were quenched with 100 mM glycine in PBS (pH 8) for 10 min. Cells were washed as above, lysed in 500 μL radioimmunoprecipitation assay buffer (RIPA) buffer, and 1 mg of cell lysate was dissolved in 500 μL of RIPA buffer supplemented with 25 μL of streptavidin agarose beads, and rotated at 4 °C for 3 h. Beads were washed 10 times in 0.5 mL RIPA buffer, boiled in sample buffer, and analyzed by sodium dodecyl sulfate-polyacrylamide gel electrophoresis (SDS–PAGE) and Western blot analysis (see [18] for the complete protocol). In these experiments, the loads used for the total protein contained 15 μg of total cell lysate. The antibodies used were: the OmicsLink™ Anti-GFP Tag Antibody Mouse Monoclonal IgG1 ((CGAB-GFP-0050); 1:1000- detects a protein band of approximately 98 kDa (HspA1A+GFP)); the beta actin antibody, Clone: 13E5, Cell Signaling ((Rabbit mAb #4970) 1:1000- detects a protein band of approximately 42 kDa); the Na+/K+ ATPase α (ATP1A1) antibody RabMAb^®^ ((EP1845Y); (2047-1); 1:1000- detects a protein band of approximately 112 kDa); and the RFP Tag Monoclonal Antibody (RF5R) ((MA5-15257); 1:1000- detects a protein band of approximately 27 kDa (RFP-C1 vector) and a protein of approximately 45kDa (Lact-C2-mCherry)). All blots were incubated with the antibodies overnight (~16 h) at 4 °C with constant rotating.

The blots were stained for total protein with the Pierce™ Reversible Protein Stain (Thermo Scientific™). The western signals were detected using either the Omega Lum C (Aplegen, San Francisco, CA, USA) or the Ci-Digit (LicoR, Lincoln, NE, USA) systems. The detected protein bands were quantified using densitometry (Image Studio Lite 3.1) and are presented as a ratio between PM (biotinylated fraction)/total (loads) protein. These experiments were repeated three times.

Furthermore, to investigate the behavior of Lact-C2 after the heat-shock treatment, we labeled with biotin (exactly as described above) cells transfected only with Lact-C2-GFP. This experiment was performed once.

Lastly, to investigate the behavior of GFP-C1-PLCdelta-PH after heat-shock and provide support to the imaging experiments, we isolated the total PM proteins from cells expressing HspA1A-RFP, and HspA1A-RFP, and GFP-C1-PLCdelta-PH. For these experiments, HEK293 cells were grown to confluency in 75 cm^2^ flasks and approximately 20 million cells were used per treatment and construct combination. The PM proteins were isolated using the Plasma Membrane Protein Extraction Kit (101Bio, Palo Alto, CA, USA) following the protocol provided by the manufacturer. This non-detergent separation generates a nuclear, a cytosolic, an internal membrane, and a plasma membrane protein fraction. Here, we present the gel of the cytosolic (60 μg protein loads) and the PM (120 μg protein loads) fractions. This experiment was performed once.

### 2.5. Statistical Tests

Statistical significance was determined by an unpaired *t*-test. A *P* value <0.05 was considered statistically significant. The boxplots were generated using R software (http://shiny.chemgrid.org/boxplotr/) [32].

## 3. Results

### 3.1. HspA1A Localizes at the PM after Heat-Shock

To determine whether GFP-tagged HspA1A translocates to the PM after mild heat-shock (42 °C for 1 h), we measured the average fluorescence of the PM-localized HspA1A in control and heat-shocked cells, which recovered for 8 h at 37 °C (Figure 1 and Appendix A). These experiments revealed that the PM localization of HspA1A increases significantly (*P* < 0.0001) by approximately 10% after heat-shock as compared to the control cells. In contrast, cytosolic EGFP alone, which was used as a negative control, did not show any significant changes in PM localization after heat-shock (Figure 1; *P* = 0.2019). Similarly, the CTCF ratio of the WGA-AF555, which was used to mark the PM of the cells, did not show any significant changes after heat-shock (*P* = 0.3151). Although the increase of HspA1A at 42 °C is significant, the CTCF ratio of HspA1A is less than half as compared to the CTCF ratio of the PM stain WGA-AF555.

### 3.2. PM Localization of HspA1A Does Not Depend on the Total Membrane Charge

To test whether HspA1A nonspecifically translocates to the PM due to purely electrostatic forces, we neutralized the total PM charge using sphingosine and determined whether this treatment alters the PM localization of HspA1A. As a positive control, we used the polybasic charge sensor R-Pre-GFP, because sphingosine inhibits its PM-localization [23]. Our results showed that the presence of sphingosine almost completely abolished the amount of R-Pre-GFP at the plasma membrane reaching basal levels. Yet, sphingosine had minimal effects on the PM-localization of HspA1A, resulting in a reduction of approximately 2% for both the control and heat-shocked conditions (Figure 2 and Appendix A). Although these changes are statistically significant, their effect size is small, suggesting that the PM-localization of HspA1A is mainly dependent on specific electrostatic interactions, and does not depend solely on the total anionic membrane charge (i.e., nonspecific electrostatic interactions).

### 3.3. PM Localization of HspA1A is Greatly Reduced by the Presence of Lact-C2

We next determined whether the PM-localization of HspA1A depends on its ability to bind to membrane lipids, such as PS. To test whether HspA1A’s PM-localization is related to binding of intracellular PS, we co-transfected HspA1A with Lact-C2, a well-established PS-biosensor, and calculated the PM-localized HspA1A (Figure 3 and Appendix A). Our results revealed that, in the presence of Lact-C2, the PM-localized HspA1A in the control cells was reduced by approximately 3%, a change that, although significant (*P* < 0.0001), had a small effect size. In contrast, in heat-shocked cells, the reduction of PM-localized HspA1A was approximately 10%, reaching the levels of the control cells (*P* < 0.0001). Similar reduction levels of PM-localized HspA1A were obtained when HEK293 or HepG2 cells were used (Appendix A). Furthermore, under our experimental conditions, the percent of apoptotic cells was very small, almost identical to the control (Appendix A); thus, PS is predominantly in the inner leaflet of the plasma membrane. These results suggest that the PM localization of HspA1A is highly dependent on PS binding [5,12,14,15,18] and this dependence is not cell-type specific.

To identify potential anionic lipid substrates in addition to PS at the plasma membrane, HspA1A-RFP was co-transfected with the phosphatidylinositol 4,5-bisphosphate (PI(4,5)P_2_) biosensor PLCδ-PH-GFP [26]. These experiments revealed that the presence of PLCδ-PH caused a small but consistent reduction of the PM-localized HspA1A in both the control and heat-shocked cells (Figure 3 and Appendix A). Although these changes are statistically significant, their effect size is relatively small when compared to the PS, suggesting that the HspA1A PM-translocation is PS-selective.

As a control, we also tested whether heat-shock or the presence of HspA1A affect the Lact-C2 or PLCδ-PH intracellular localization. These analyses, which are supported by biochemical experiments, revealed that the localization of both Lact-C2 and PLCδ-PH remain largely unaffected by the treatments (Figure 5 and Appendix A). Furthermore, these analyses revealed that the CTCF ratio (mean value = 0.31) of HspA1A at 42°C is comparable although significantly lower than the Lact-C2 (mean value = 0.35; *P* = 0.0028) and PLCδ-PH (mean value = 0.37; *P* < 0.0001) ratios under the same conditions.

### 3.4. HspA1A Co-Localizes with Intracellular PS

Taking into account that HspA1A’s PM localization depends on its binding to intracellular PS, we sought to test whether the protein co-localizes with TopFluor-PS, a fluorescent PS analog that is not metabolized [33], at 37 °C. For these sets of experiments, we used Lact-C2 as a positive control and our results in agreement with the literature [33] showed that Lact-C2 has a high level of co-localization with PS at the PM and most probably in endosomes (Mander’s coefficient = 0.88) (Figure 4 and Appendix A). In the case of HspA1A, our results revealed that the chaperone co-localizes with PS primarily at the PM (Mander’s coefficient = 0.80) (Figure 4). This level of co-localization, although lower than the one observed for Lact-C2, is significantly higher than expected by chance, suggesting that HspA1A binds to intracellular PS.

### 3.5. HspA1A Embeds at the PM after Heat-Shock and This Ability is PS-Selective

We next sought to verify the imaging results using cell surface biotinylation. In these experiments, we co-transfected GFP-tagged HspA1A with an empty mRFP-C1 vector for two reasons. First, to control for nonspecific effects on the PM localization of HspA1A due to the presence of any fluorescent protein and second, to equalize the amount of transfected DNA between the different experiments. We also co-transfected GFP-HspA1A with mCherry-Lact-C2 to verify that the presence of Lact-C2 inhibits the PM-localization of HspA1A. These experiments revealed that HspA1A’s PM localization is increased at 8 h of recovery after heat-shock as compared to control cells (Figure 5) and verified the imaging results (see Figure 1). Furthermore, the PM-localized HspA1A was only subtly altered in the presence of sphingosine, while it was almost diminished in the presence of Lact-C2 (Figure 5 and Appendix A). Furthermore, analysis of isolated total PM protein fractions suggests that the presence of PLCδ-PH has minimal effects on the PM localization of HspA1A (Appendix A). These results verify the imaging experiments (Figure 1, Figure 2 and Figure 3), strongly suggesting that HspA1A embeds within the lipid bilayer, and establish that PS-selectivity is a critical factor for the PM-localization and anchorage of HspA1A.

## 4. Discussion

In this study, we determined that the PM-localization of the cytosolic molecular chaperone HspA1A depends on its binding to intracellular PS. Five key findings support our conclusion. (A) In three different cell lines, after mild heat-shock and during recovery, HspA1A translocated and embedded to the PM (Figure 1, Figure 2 and Figure 3 and Appendix A). (B) PM-localization of HspA1A was affected minimally after neutralization of the membrane charge (Figure 2 and ). (C) HspA1A PM-localization and embedding was abolished by removal of available PS targets by co-transfection with the PS-biosensor Lact-C2 [18,34] (Figure 2 and Figure 3 and Figure 5, and Appendix A). (D) HspA1A co-localized with fluorescently labeled PS (Figure 4). (E) HspA1A PM-localization was only slightly reduced after removal of PI(4,5)P_2_ (Figure 3 and Appendix A).

Our findings on HspA1A extend the current understanding of this new, yet controversial, lipid-binding function of HspA1A, and imply that HspA1A PM-localization and anchorage is not cell-type specific and is minimally related to the negative charge of the PM and its anionic lipids. Furthermore, our findings show that HspA1A binds to intracellular PS, because the mild heat-shock used did not cause apoptosis, and thus PS was largely contained in the inner leaflet of the PM. Additionally, our results suggest that although electrostatic interactions, for example, hydrogen bonds between HspA1A and PI(4,5)P_2_ headgroups, are not sufficient to explain the membrane binding of HspA1A, they are still important [18,19]. Lastly, our findings strongly support the notion that the selective binding between HspA1A and intracellular PS is important for the stress-induced membrane localization and embedding of HspA1A.

In agreement with previous studies [2,3,10,35,36], our results solidify the notion that HspA1A translocates and anchors to the PM of stressed and cancer cells [7,9,16,17], making it a conserved property of this important molecular chaperone. Furthermore, our experiments strongly support the idea that HspA1A binds to PS and other lipids. In the case of PS in particular, previous in vitro studies, using purified recombinant HspA1A and artificial liposome membranes, have showed that HspA1A binds to PS with higher selectivity and affinity as compared to other anionic lipids [5,7,11,12,13,14,15,16,17,18,19]. For example, HspA1A binds to PS with 10 times higher affinities as compared to other lipids, such as Bis(monoacylglycerol)phosphate, phosphatidic acid, and several phosphoinositides [18,19]. The interaction of HspA1A with lipids has also been showed ex vivo. In particular, HspA1A was found in lipid rafts, which are membrane domains enriched in cholesterol and sphingolipids, such as ganglioside 1 (GM1) and Gb3 [4,7,16,36,37,38]. Phospholipids, such as PS, are present in small amounts; however, recent research shows that in caveolae, a subset of lipid rafts, PS plays a crucial role in both caveolae formation and stability [39]. In lipid rafts, HspA1A was found to bind Gb3, which exists in the outer leaflet of the PM [7,40]. Furthermore, HspA1A was found to bind to extracellularly localized PS in pre-apoptotic cells [5,16]. Our findings, in addition to providing strong ex vivo support to the lipid-binding function of HspA1A, advance our understanding of this new property of HspA1A, because they reveal that HspA1A binds to intracellularly localized PS. Furthermore, our results support the notion that the binding of HspA1A to PS and other lipids is physiologically important. For example, binding of HspA1A to intracellular PS or extracellular Gb3 plays a critical role in the localization of HspA1A at the PM of stressed and tumor cells, while binding to extracellularly localized PS regulates cell survival [5,16].

Collectively, our experiments strongly support the notion that selective binding of HspA1A to PS plays a critical role in the localization and anchorage of the chaperone to the PM. However, they do not establish PS specificity, because the effect of binding to phosphoinositide phosphates other than PI(4,5)P_2_ has not been established. Additional experiments using lipid-biosensors for other negatively charged lipids as well as different pharmacological manipulations are required to determine whether binding to PS alone is sufficient for HspA1A to localize at the PM or other negatively charged lipids are also important. Furthermore, our results using biotin labeling strongly suggest that HspA1A fully integrates into the PM and even partially translocates through the PM. These results are strongly supported by several reports showing that HspA1A embeds within the PM of stressed cells [7,9,17]. An alternative interpretation of the biotinylation results, which would be that HspA1A is secreted and then labeled by biotin, although cannot be formally ruled out, is less plausible given the experimental protocol used, the Lact-C2 biotin experiments, and the timing of secretion [17]. Although these combined observations strongly imply that HspA1A spans the PM, leaving a region of the protein in the extracellular space, they do not clarify why the embedding is highly increased in stressed and cancer cells [7,9,17]. Nevertheless, taking into account several reports and observations, we speculate that the HspA1A’s PM-embedding is promoted by the increase of the PS saturation levels [20]. This speculation is supported by three major findings. First, the amount of PS accessible to Lact-C2 is not significantly increased by heat-shock (similarly, the amount of PS in cancer cells does not increase [7,41]. Second, HspA1A embeds selectively into artificial membranes composed only of saturated PS [12,15,20]. Third, both heat-shocked and cancer cells are enriched with saturated lipids [42,43,44,45].

The interaction of HspA1A (and other molecular chaperones) with cellular and organelle membranes and lipids has received a lot of attention in recent years because of the chaperones’ association with cancer. Although multiple studies have revealed important information regarding the mechanism and biological implications of the interaction between HspA1A and membrane lipids, particular mechanistic details of the pathway remain to be elucidated. For example, it is not clear whether this phenomenon is directly related to modifications of the lipid species, e.g., increased saturation levels [42,43,44,45] and it is unclear whether HspA1A’s PM localization is directly related to its secretion. Furthermore, the importance of other lipids, which HspA1A is known to bind with relatively high affinity, e.g., Phosphatidylinositol 4-phosphate (PI(4)P), is yet to be determined. The identification and characterization of the molecular mechanism that drives the chaperone to the PM and results in its secretion is a critical step towards understanding the physiological significance of this unconventional property of HspA1A.

## 5. Conclusions

In conclusion, our study shows that HspA1A translocates at the PM of human cell lines during recovery from mild non-apoptotic heat-shock. Furthermore, it shows that selective binding to intracellular PS enables HspA1A’s PM-localization and embedding. Collectively, these findings demonstrate the importance of the interaction between HspA1A and PS in non-apoptotic cells for the protein’s PM-localization, and further solidify the biological significance of HspA1A–lipid interactions, which have important implications in cancer biology [5,7,9,12,15,16,17,18,20].

## Figures and Tables

**Figure 1 biomolecules-09-00152-f001:**
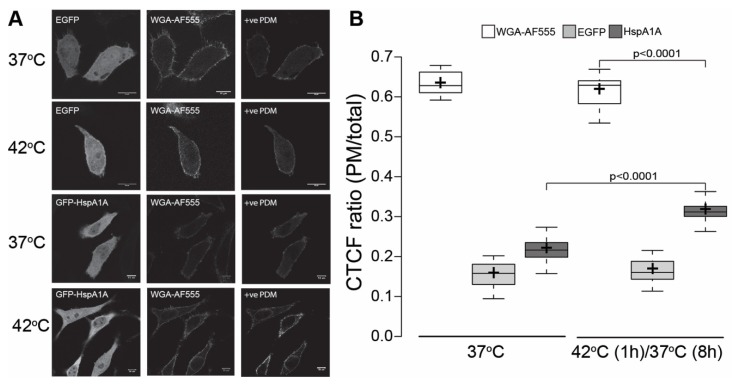
Plasma membrane localization of HspA1A increases during recovery from mild heat-shock at 42 °C. (**A**) Representative images of Henrietta Lacks’ ‘Immortal’ (HeLa) cells expressing empty EGFP or GFP-HspA1A. The cells were grown under normal conditions (37 °C) or heat-shocked for 1 h at 42 °C and allowed to recover for 8 h. The plasma membrane was stained using the wheat germ agglutinin lectin Alexa Fluor^®^ 555 conjugate (WGA-AF555). The images showing the +ve Product of the Differences from the Mean (PDM) (see materials and methods) values were included to show co-localization. Scale bar = 10 μm. (**B**) Quantification of the corrected total cell fluorescence (CTCF) as a ratio between the total GFP-HspA1A fluorescence of the plasma membrane and the rest of the cell. Center lines show the medians; box limits indicate the 25th and 75th percentiles as determined by R software; whiskers extend 1.5 times the interquartile range from the 25th and 75th percentiles; crosses represent sample means. The experiment was repeated three times and the number of cells used to generate each boxplot was (from left to right) *n* = 16, 34, 48, 17, 38, and 41. The *P* values of the student *t*-test were: WGA-AF555-42/HspA1A-GFP-42 < 0.0001; GFP-HspA1A-37/GFP-HspA1A-42 < 0.0001; GFP-37/GFP-42 = 0.2019; GFP-37/GFP-HspA1A-37 < 0.0001; GFP-42/GFP-HspA1A-42 < 0.0001.

**Figure 2 biomolecules-09-00152-f002:**
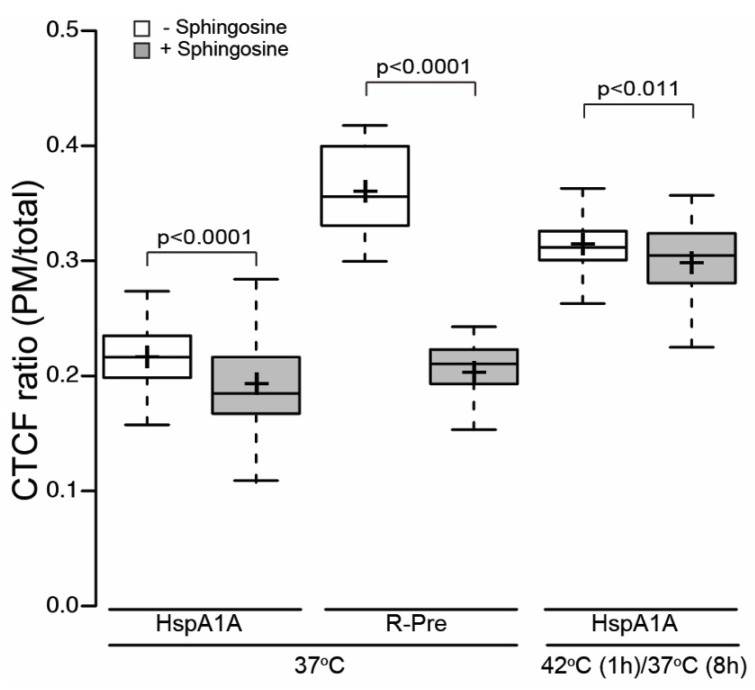
The plasma membrane (PM)-localization of HspA1A is minimally affected by the total membrane charge. Quantification of the corrected total cell fluorescence (CTCF) as a ratio between the total GFP-HspA1A fluorescence of the plasma membrane and the rest of the cell in the presence or absence of sphingosine. The polybasic charge sensor R-Pre-GFP was used as a control. Center lines show the medians; box limits indicate the 25th and 75th percentiles as determined by R software; whiskers extend 1.5 times the interquartile range from the 25th and 75th percentiles; crosses represent sample means. The experiment was repeated three times and the number of cells used to generate each boxplot was (from left to right) *n* = 48, 32, 28, 26, 41, 39.

**Figure 3 biomolecules-09-00152-f003:**
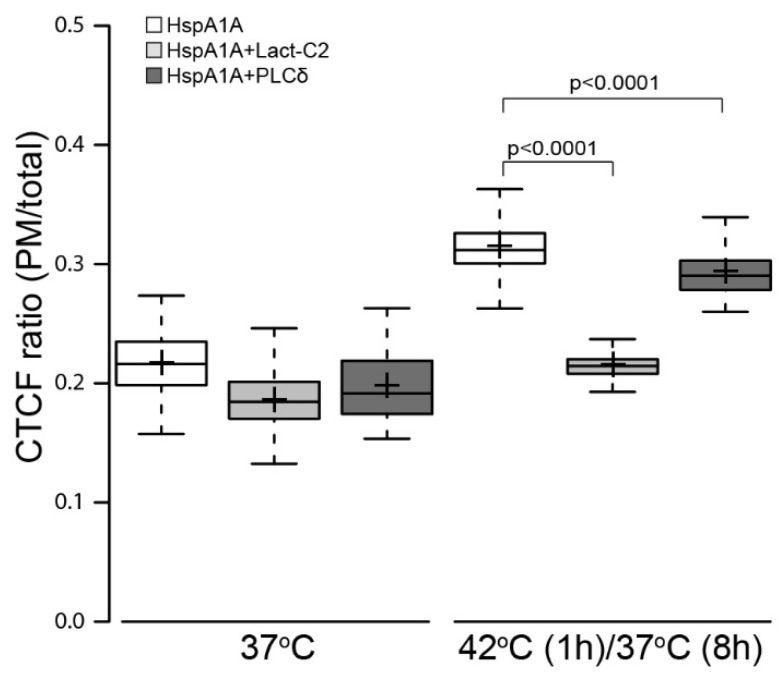
The PM-localization of HspA1A is inhibited by the removal of phosphatidylserine (PS) (presence of Lact-C2), but it is minimally affected by the removal of PI(4,5)P_2_ (presence of PLC-δ-PH). The quantification of the corrected total cell fluorescence (CTCF) as a ratio between the total GFP-HspA1A fluorescence of the plasma membrane and the rest of the cell is presented for the different conditions used. Center lines show the medians; box limits indicate the 25th and 75th percentiles as determined by R software; whiskers extend 1.5 times the interquartile range from the 25th and 75th percentiles; crosses represent sample means. The experiment was repeated three times and the number of cells used to generate each boxplot was (from left to right) *n* = 48, 44, 26, 41, 52, 34.

**Figure 4 biomolecules-09-00152-f004:**
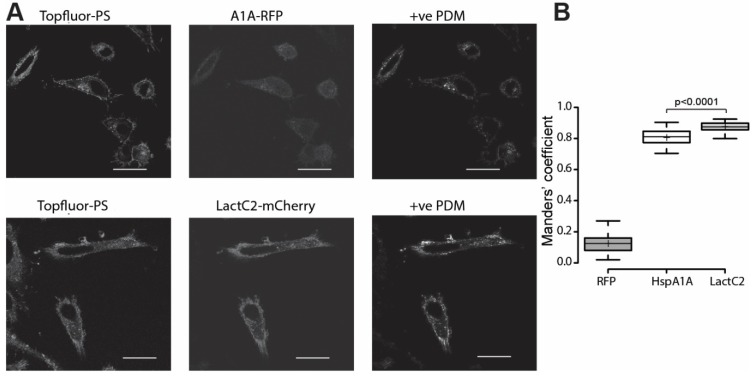
HspA1A co-localizes with TopFluor-PS in human cells. (**A**) Representative images of HeLa cells showing the co-localization of HspA1A (top panel) and Lact-C2 (lower panel) with PS at 37 °C. (**B**) Quantification of the co-localization using the Manders’ co-localization coefficient analyses using 20, 26, and 25 cells from three independent experiments. Center lines show the medians; box limits indicate the 25th and 75th percentiles as determined by R software; whiskers extend 1.5 times the interquartile range from the 25th and 75th percentiles; crosses represent sample means. The images showing the +ve PDM (see materials and methods) values were included to show co-localization. Scale bar = 10 μm. The *P* values of the student *t*-test were: RFP/HspA1A < 0.0001; RFP/ LactC2 < 0.0001; HspA1A/LactC2 < 0.0001.

**Figure 5 biomolecules-09-00152-f005:**
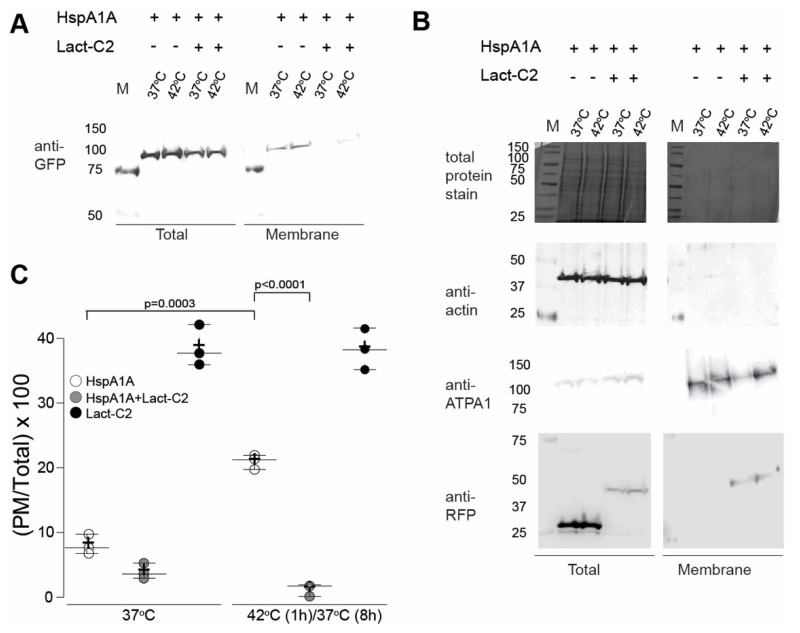
Cell surface biotinylation reveals that HspA1A’s PM-localization depends on its binding to PS. (**A**) Representative Western blots showing the total and biotinylated fractions of HEK293 cell lysates transfected with HspA1A-GFP and RFP-C1 vector, and HspA1A-GFP and Lact-C2-mCherry. The antibody used was the OmicsLink™ Anti-GFP Tag Antibody Mouse Monoclonal IgG1 ((CGAB-GFP-0050); 1:1000); (**B**) The nitrocellulose membranes used in (**A**) were stained with the Reversible Protein Stain Kit (Pierce) and were also blotted with the control antibodies listed below. The antibodies used were the Beta Actin, Clone: 13E5, Cell Signaling ((Rabbit mAb #4970) 1:1000); Na+/K+ ATPase α (ATP1A1) antibody RabMAb^®^ ((EP1845Y); (2047-1); 1:1000); and RFP Tag Monoclonal Antibody (RF5R) ((MA5-15257); 1:1000)). M: molecular size marker (BioRad Dual Color Protein Ladder; approximate sizes shown on the left side of the blots). (**C**) Quantification of the antibody detected signals of the HspA1A (GFP tagged) in the presence or absence of Lact-C2, as well as the Lact-C2 (mCherry tagged) presented as a ratio between the biotinylated (PM) fraction and the total cell lysate. Densitometry values are averages of three independent experiments (*n* = 3). Center lines show the medians; whiskers extend 1.5 times the interquartile range from the 25th and 75th percentiles; crosses represent sample means. The *P* values of the student *t*-test were: HspA1A-37/HspA1A-42 = 0.003; HspA1A-42/HspA1A-42-LactC2 < 0.0001; HspA1A-37-LactC2/HspA1A-42-LactC2 = 0.0413; HspA1A-37/HspA1A-37-LactC2 = 0.0212.

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
