# Peer review of "Membrane Localization of HspA1A, a Stress Inducible 70-kDa Heat-Shock Protein, Depends on Its Interaction with Intracellular Phosphatidylserine"

_biomolecules, 2019, doi:10.3390/biom9040152_

Round 1

Reviewer 1 Report

This paper describes the plasma membrane (p.m.) localization of the major Hsp70 HspA1A in response to heat shock. Furthermore, the paper shows evidence for the mechanism of p.m. interaction, namely through specific binding to the lipid PS. For the most part, the manuscript is clearly written, the experiments careful designed and the results are well presented. That said, I do believe there are some major caveats to the paper’s conclusion that HspA1A is binding PS selectively; specifically, the authors should address the following:

PS is present in high concentration at the p.m. (and endosomes) of cells under non-stressed conditions - so why does HspA1A show translocation to the p.m. (and endosomes) specifically after heat shock?

The evidence for PS selectivity comes from the failure of sphingosine to displace HspA1A, ruling out electrostatics. However, the protein could be interacting with a number of other membrane lipids, including assorted phosphoinositides; this could be tested with generic inhibitors such as phenylarsine oxide or wortmannin at micromolar concentrations.

The “sequestration” of PS by Lact-C2 is unlikely to be the explanation for the observed decrease in localization. PS  is present at far higher concentrations in cells (equivalent to several mM if it were dissolved in the cytosol), whereas the most abundant over-expressed proteins for protein purification purposes struggle to get above tens of micromolar. Notably, there is no control for displacement of non-PS dependent peripheral membrane proteins in this context. Dependence on PS could be alternatively tested by activating PS scrambling with calcium ionophore (though it would need the above controls in point 2 to rule out effects on phosphoinositides).

Minor comments:

The contrast on the images in all figures showing the red channels is not sufficient and needs to be enhanced for clarity.

t-tests are used throughout the paper, though multiple comparisons are being made and there is apparently no correction of the p value. The authros must run suitable tests for multiple comparisons in each experiments, such as one- or two-way ANOVA instead with appropriate post-hoc tests.

The methods describes mounting the cells in Fluoromount-G. This is an aqueous/g

Author Response

Reviewer 1

  Yes

Can be improved

Must be improved

Not applicable

Does the introduction provide sufficient   background and include all relevant references?

(x)

( )

( )

( )

Is the research design appropriate?

(x)

( )

( )

( )

Are the methods adequately described?

(x)

( )

( )

( )

Are the results clearly presented?

( )

( )

(x)

( )

Are the conclusions supported by the results?

( )

( )

(x)

( )

Comments and Suggestions for Authors

This paper describes the plasma membrane (p.m.) localization of the major Hsp70 HspA1A in response to heat shock. Furthermore, the paper shows evidence for the mechanism of p.m. interaction, namely through specific binding to the lipid PS. For the most part, the manuscript is clearly written, the experiments careful designed and the results are well presented. That said, I do believe there are some major caveats to the paper’s conclusion that HspA1A is binding PS selectively; specifically, the authors should address the following:

We would like to thank the reviewer for taking the time to read our manuscript and providing very timely and critical feedback in improving its quality.

PS is present in high concentration at the p.m. (and endosomes) of cells under non-stressed conditions - so why does HspA1A show translocation to the p.m. (and endosomes) specifically after heat shock?

Response: We agree with the reviewer that this is an important question; however, we do not know why HspA1A translocates at the PM during recovery from heat shock. This very important question is part of our ongoing investigation to determine the molecular mechanism of the HspA1A PM-localization. Although are important, these experiments are out of the scope of this manuscript, which was to determine whether HspA1A binds to intracellular PS and whether this binding is related to its PM localization. Based on the current evidence and our findings that PM localized Lact-C2 does not change significantly after heat-shock (Figure 5 and Supplementary Figure 5), we speculate that the increase in PM localization of HspA1A after heat-shock might be related to the increase of saturation of PS after this stress. The speculation is indirectly supported by different reports revealing that HspA1A selectively embeds in liposomes composed of saturated PS (Armijo et al. Cell Stress Chaperones 2014; McCallister et al. Cell Stress Chaperones 2015).

The evidence for PS selectivity comes from the failure of sphingosine to displace HspA1A, ruling out electrostatics. However, the protein could be interacting with a number of other membrane lipids, including assorted phosphoinositides; this could be tested with generic inhibitors such as phenylarsine oxide or wortmannin at micromolar concentrations.

Response: This is an excellent point. We agree with the reviewer that PS might not be the only lipid responsible for the PM localization of HspA1A. Others and we have showed that HspA1A binds to different lipids [including phospho-phosphoinositides (PIPs) with varied affinities in vitro (McCallister et al. Biochem Biophys Res Commun 2015; McCallister et al. Cell Stress Chaperones 2015; McCallister et al. Sci Rep 2015; McCallister et al. Biochem Biophys Res Commun 2016). Although the experiments using particular PPIs inhibitors are important, they are out of the scope of the current manuscript, which was focused on PS. We did, however, use the PLC-delta-PH domain as a control to suggest selectivity for PS. Notably, the PLC-delta-PH domain is selective for PI(4,5)P2 and as shown in Figure 3 PLC-delta-PH does have a small but statistically significant effect on HspA1A localization after heat shock.  This finding is now also discussed in more detail in the revised manuscript.

The “sequestration” of PS by Lact-C2 is unlikely to be the explanation for the observed decrease in localization. PS is present at far higher concentrations in cells (equivalent to several mM if it were dissolved in the cytosol), whereas the most abundant over-expressed proteins for protein purification purposes struggle to get above tens of micromolar.

Response: We agree with the reviewer that Lact-C2 may not selectively remove all available PS from the cell. Based on reports from Grinstein’s group PS ranges from 100-300μM and an overexpressed protein could be around 25μM (Botelho et al 2000-JCB, 151,1353-1367; Kay et al. 2011-Sensors, 11:1744-1755; Kay et al. 2012-Mol Biol Cell, 23:2198-2212). However, our data show that even this amount of PS removal alters the PM localization of HspA1A. Further, while the Lact-C2 concentration may be in the 20-30 μM range, protein binding to membrane can cover more than one lipid (the N number) and thus more than one PS molecule is likely to be blocked per molecule of Lact-C2.

Notably, there is no control for displacement of non-PS dependent peripheral membrane proteins in this context.

Response: We used the PLC-delta-PH domain as a control to suggest selectivity for PS.  Please see Figure 3 and Supplementary Figure 6.

Dependence on PS could be alternatively tested by activating PS scrambling with calcium ionophore (though it would need the above controls in point 2 to rule out effects on phosphoinositides).

Response: Similarly, to the previous comment we believe that using ionomycin is a great idea. However, it is out of the scope of the current manuscript.  Raising the intracellular calcium will also increase phospholipase activity, leading to phosphoinostitide headgroup cleavage. Thus, while PS exposure to the outer leaflet may increase, the pool of PIPs at the inner leaflet of the plasma membrane would decrease, making it more difficult to interpret the role of PS vs PIPs.

Minor comments:

The contrast on the images in all figures showing the red channels is not sufficient and needs to be enhanced for clarity.

Response: We corrected this issue with the images presented in the revised manuscript.

t-tests are used throughout the paper, though multiple comparisons are being made and there is apparently no correction of the p value. The authros must run suitable tests for multiple comparisons in each experiments, such as one- or two-way ANOVA instead with appropriate post-hoc tests.

Response: We compared the means of two samples each time, and for these comparisons, we applied the t-test. We did not perform multiple (dependent or independent) statistical tests simultaneously; therefore, correcting for multiple testing (Bonferroni or other family-wise error rate control procedure) is not applicable in our study. Additionally, analysis of variance (ANOVA) is not suitable in our case, because we did not compare the means of several groups simultaneously.

The methods describes mounting the cells in Fluoromount-G. This is an aqueous/g

Response: We did use Fluoromount-G, which is a water-soluble compound, but in order to be removed it requires a few hours of submerging in 1XPBS solution. The mounting medium provides a semi-permanent seal for storage of slide preparations.   

Reviewer 2 Report

The manuscript entitled “Membrane localization of HspA1A, a stress inducible 70-kDa heat shock protein, depends on its interaction with intracellular phosphatidylserine” submitted by Nikolas Nikolaidis and colleagues shows that the chaperone HpsA1A interacts with phsophatdylserine (PS) to allow its recruitment at the plasma membrane.
This study is interesting as it shows in vivo data allowing to better understand the cellular role of the PS-binding by Hps70. However, some important controls are missing and the quality of the fluorescent images of the transfected cells is very weak for some experiments, this has to be improved.

Comments:

1. The WGA-AF555 wheat germ agglutinin is used to label the cell surface. This should be indicated into the Figure legends.

2. The WGA-AF555 staining of the cell surface is very weak in Figure 1, and not homogeneous, as it appears at a dotted staining of the plasma membrane. Moreover, the WGA signal is almost not detectable in Sup Fig 1 (panel EGFP and GFP-HspA1A 37°C) and in Sup Fig 2B (panel 37°C +SP). This point needs to be clarified as this WGA-AF555 staining is used to determine HspA1A localization beneath the plasma membrane.

4. Although the results are clear upon transfection with the mCherry-LactC2 probe, the quality of the microscopy images is very weak, as shown in Supplementary Fig. S2, and the transfected cells look damaged; having a DAPI staining showing the nucleus would be helpful. Moreover, the mCherry-LactC2 probe should be localized beneath the plasma membrane, whereas its localization seems to be more next to the nucleus (Supplementary Fig. 2) with a pattern corresponding to Golgi network, this could be ascertain by using a Golgi probe (as GM130 for example). The control experiment showing the mCherry-LactC2 transfected cells co-transfected with GFP is also missing, and this control is critical to show that co-expression of HpsA1A did not alter the localization of the mCherry-LactC2 probe at the PS-enriched plasma membrane sites. This control is also essential to decipher the localization of mCherry-LactC2 upon treatment at 42°C, as this treatment might change the PS localization or turn-over.

5. The authors claim that the PM localization of HspA1A-RFP is reduced upon expression of the PLC-PH-GFP biosensor, but I do not see any fluorescent signal corresponding to the expression of HspA1A-RFP, this imaging has to be improved. The authors also have to show that this fusion construct is properly expressed in the transfected cells by doing a western-blot anti-RFP.

Author Response

Reviewer 2

Yes

Can be improved

Must be improved

Not applicable

Does the introduction provide sufficient   background and include all relevant references?

(x)

( )

( )

( )

Is the research design appropriate?

( )

( )

(x)

( )

Are the methods adequately described?

(x)

( )

( )

( )

Are the results clearly presented?

( )

(x)

( )

( )

Are the conclusions supported by the results?

( )

(x)

( )

( )

Comments and Suggestions for Authors

The manuscript entitled “Membrane localization of HspA1A, a stress inducible 70-kDa heat shock protein, depends on its interaction with intracellular phosphatidylserine” submitted by Nikolas Nikolaidis and colleagues shows that the chaperone HpsA1A interacts with phsophatdylserine (PS) to allow its recruitment at the plasma membrane. 
This study is interesting as it shows in vivo data allowing to better understand the cellular role of the PS-binding by Hps70. However, some important controls are missing and the quality of the fluorescent images of the transfected cells is very weak for some experiments, this has to be improved.

We would like to thank the reviewer for taking the time to read our manuscript and providing very timely and critical feedback in improving its quality.

Comments:

1. The WGA-AF555 wheat germ agglutinin is used to label the cell surface. This should be indicated into the Figure legends.

Response: We apologize for the omission in the original manuscript. We added this experimental detail in the legends of the revised manuscript.

2. The WGA-AF555 staining of the cell surface is very weak in Figure 1, and not homogeneous, as it appears at a dotted staining of the plasma membrane. Moreover, the WGA signal is almost not detectable in Sup Fig 1 (panel EGFP and GFP-HspA1A 37°C) and in Sup Fig 2B (panel 37°C +SP). This point needs to be clarified as this WGA-AF555 staining is used to determine HspA1A localization beneath the plasma membrane.

Response: The WGA stain used is a lectin binding to sialic acid and N-acetylglucosaminyl residues (Invitrogen). We corrected the contrast of the images to make them clearer (or changed the images presented). The stain delivers a spotty PM appearance (please see image below from Invitrogen’s website) and the pictures we obtained have this appearance. Additionally, the cell (and outer plasma membrane) is a three-dimensional object and each image is taken in a single z-plan, this can often explain the spotty or different appearance of the WGA stain.

We also need to apologize for not making clear in the original submission what was the purpose of using the PM stain, which was only to mark the PM of the cells and not to quantify HspA1A at the PM. We now clarify the reasoning of using the lectin stain in the revised manuscript.

4. Although the results are clear upon transfection with the mCherry-LactC2 probe, the quality of the microscopy images is very weak, as shown in Supplementary Fig. S2, and the transfected cells look damaged; having a DAPI staining showing the nucleus would be helpful.

Response: Following the reviewer’s suggestion, we have included the DAPI in the revised figures. These images do not show any fragmentation of the nucleus. Additionally, we would like to point out that the cells under our treatments are not pre-apoptotic (see Supplementary Figure 4).

Moreover, the mCherry-LactC2 probe should be localized beneath the plasma membrane, whereas its localization seems to be more next to the nucleus (Supplementary Fig. 2) with a pattern corresponding to Golgi network, this could be ascertain by using a Golgi probe (as GM130 for example).

Response: The subcellular localization of Lact-C2 (please see figure below from relevant literature: Yeung et al. 2008; Science 319:210-213) has been described in detail. According to their results Lact-C2 (unexpectedly) does not significantly co-localizes with the Golgi. While Lact-C2 was not found to be significantly localized to the Golgi, mitochondria, or ER (Yeung et al. 2008 as well as below), Lact-C2 could bind the cytosolic face of vesicles of the endosomal pathway. Furthermore, comparing these pictures to the ones we obtained, it is evident that the apparent localization of Lact-C2 appears different depending on the cell type and cell morphology.

The control experiment showing the mCherry-LactC2 transfected cells co-transfected with GFP is also missing, and this control is critical to show that co-expression of HpsA1A did not alter the localization of the mCherry-LactC2 probe at the PS-enriched plasma membrane sites. This control is also essential to decipher the localization of mCherry-LactC2 upon treatment at 42°C, as this treatment might change the PS localization or turn-over.

Response: Taking into account this excellent suggestion, we analyzed further the PM localization of Lact-C2 in the presence or absence of HspA1A (tagged with GFP) at 37oC and 42oC. We present these data in Supplementary Figure 5 in the revised manuscript. Based on these results we propose that the localization of Lact-C2 seems to be unaffected from either the stress or the presence of HspA1A. In the discussion, we suggest that at least the amount of PS being sequestered by Lact-C2 is also unaffected by the heat-treatment. In the revised text, we also include similar analysis for the PLC-PH domain (PI(4,5)P2 binding).

5. The authors claim that the PM localization of HspA1A-RFP is reduced upon expression of the PLC-PH-GFP biosensor, but I do not see any fluorescent signal corresponding to the expression of HspA1A-RFP, this imaging has to be improved. The authors also have to show that this fusion construct is properly expressed in the transfected cells by doing a western-blot anti-RFP.

Response: We apologize for this issue. We corrected the contrast of the images to make them clearer. We also show a blot using the RFP-HSPA1A (Supplementary Figure 6).

Reviewer 3 Report

In Bilog et they describe the translocation of the chaperone HspA1 from the cytosol to the plasma membrane during the recovery phase from non apoptotic heat-shock. They show that this recolocalization is based on the selective binding of HspA1A to phosphatydilserin (PS). The whole study is based on the quantification of light microscopy micrographs applying what the authors called Product of the Difference of the Mean (PDM) just to compare the difference fluorescence intensity inside the cell and at the plasma membrane.

The paper show several flaws that should addressed before being considered for publication:

In figure 1 the stiaining with the lectin WGA, which is not mentioned in the text as a PM marker, is too dim and it is difficult to see. Related to this, the supplementary is redundant: both section A and B are just a pseudocoloured copied imaged from section A in figure 1.

Looking at the plot in figure 2 it difficult to explain why the difference shown at 42ºC between the condition with or without sphingosine is significant but not 37ºC. On the other hand the conclusion the author reach from this experiment is confusing and should be considered other effect of overloading a membrane with sphingosine apart from the change in the charge.

The images showing the colocalization of HspA1A whit the fluorescent PS analog, TopFluor-PS, don’t show clearly the colocalization. I am also surprised by the fact that the experiment has only been made at 37ºC, since it was expecting to do the experiment during the recovery after the heat shock at 42ºC, when the translocation of the chaperone to the PM is more conspicuous.

 Figure 5 seems to be OK on the screen, but in the printed version of the pdf file, the contrast/grey levels of the blot in section C changes completely and band can be seen in all the tested conditions, even in the presence of Lact-C2.

Author Response

Reviewer 3

Yes

Can be improved

Must be improved

Not applicable

Does the introduction   provide sufficient background and include all relevant references?

( )

(x)

( )

( )

Is the research design   appropriate?

( )

( )

(x)

( )

Are the methods   adequately described?

(x)

( )

( )

( )

Are the results   clearly presented?

( )

( )

(x)

( )

Are the conclusions   supported by the results?

( )

( )

(x)

( )

Comments and Suggestions for Authors

In Bilog et they describe the translocation of the chaperone HspA1 from the cytosol to the plasma membrane during the recovery phase from non apoptotic heat-shock. They show that this recolocalization is based on the selective binding of HspA1A to phosphatydilserin (PS). The whole study is based on the quantification of light microscopy micrographs applying what the authors called Product of the Difference of the Mean (PDM) just to compare the difference fluorescence intensity inside the cell and at the plasma membrane.

We would like to thank the reviewer for taking the time to read our manuscript and providing very timely and critical feedback in improving its quality.

Response: We believe that this is a misunderstanding and we apologize for not being clearer in the initial submission. In the revised text, we clarify these details. Specifically, we further describe the corrected total cell fluorescence (CTCF; described in Burgess et al. 2010-Proc Natl Acad Sci USA 107:12564–12569), which was used to determine the PM localization of HspA1A. Additionally, we clarify that we did not use colocalization or Intensity Correlation Analysis (Li et al. 2004-J Neuroscience 24:4070-4081) to quantify the changes in HspA1A’s PM localization. We also provide further description on the +ve PDM values. These are incorporated in the figures for two reasons. First, they are clear to see in the black and white color scheme. Second, they provide a quantitative idea on the correlation than image superimposition alone.

The paper show several flaws that should addressed before being considered for publication:

In figure 1 the stiaining with the lectin WGA, which is not mentioned in the text as a PM marker, is too dim and it is difficult to see. Related to this, the supplementary is redundant: both section A and B are just a pseudocoloured copied imaged from section A in figure 1.

Response: We apologize for these issues. In the initial submission, we included the WGA staining in the methods section. In the revised manuscript, we also included this information in the figure legends. We also further clarify and explain the purpose of the lectin stain in the experiments. We also changed the contrast (or the pictures) for easier visualization.

Looking at the plot in figure 2 it difficult to explain why the difference shown at 42ºC between the condition with or without sphingosine is significant but not 37ºC.

Response: We apologize for this omission. We now show the p-value in the revised figure.

On the other hand the conclusion the author reach from this experiment is confusing and should be considered other effect of overloading a membrane with sphingosine apart from the change in the charge.

Response: In this experiment, we used R-Pre as a control, because the presence of sphingosine should remove most of this marker from the PM. We also performed the experiment with HspA1A and found a very small amount of change. Based on these two results our conclusion was that electrostatics alone could not explain the presence of HspA1A at the PM. We believe that this is a valid suggestion based on the data and we are not sure what other interpretation is possible.

The images showing the colocalization of HspA1A whit the fluorescent PS analog, TopFluor-PS, don’t show clearly the colocalization.

Response: We used intensity correlation analysis (showed as PDM values) and we quantified the extent of co-localization using the Mander’s coefficient to reveal the co-localization. In the revised manuscript, we provide a Supplementary Figure 7 showing how these data look in color aiming to explain better the results.

I am also surprised by the fact that the experiment has only been made at 37ºC, since it was expecting to do the experiment during the recovery after the heat shock at 42ºC, when the translocation of the chaperone to the PM is more conspicuous.

Response: The experiment was performed only at 37ºC because TopFluor-PS is a non-metabolized analog of endogenous PS and it cannot likely be modified by heat-shock. Additionally, HspA1A is at the PM of these cells at 37ºC and it should bind to PS.

Figure 5 seems to be OK on the screen, but in the printed version of the pdf file, the contrast/grey levels of the blot in section C changes completely and band can be seen in all the tested conditions, even in the presence of Lact-C2.

Response: We thank the reviewer for pointing this out. Sometimes resolution can be lost in printing depending on the computer, printer driver, and type of printer. We also agree with the reviewer’s interpretation that HspA1A is always present at the PM (a phenomenon that might related to the cancer ancestry of the cells used). However, the amount of the protein at the PM is increasing after heat-shock and does not increase in the presence of Lact-C2. Furthermore, the quantification of the data shows that the ratio PM/Cyt are not zero.

Reviewer 4 Report

This manuscript aims to provide evidence supporting the notion that the heat shock protein HSPA1A may associate with the plasma membrane lipid phosphotidylserine (PS) in a temperature-dependent manner. The majority of the data presented is based on confocal image analyses,  which is complimented with limited supporting biochemical evidence toward the end of the Results section. Although the overall design of the experiments appears to be reasonable, this manuscript lacks sufficient detail and control experiment in support of the rationale for their image analysis protocols and data interpretation. Moreover, for surface biotinylation analyses, a significant  discrepancy was observed that raises serious concern about the quality of their biochemical data. Listed below are some of my major concerns regarding this manuscript.

1)   Given that virtually all the evidence is based on confocal image analyses, the authors are obligated to provide a comprehensive description of their image acquisition and analysis protocols in the Methods section, which however is lacking. For example, regarding the definition and quantification of the so-called plasma membrane (PM) pixels, readers are expected to  look it up in a previous publication (ref. 29) by one of the co-authors that addresses the membrane lipid-association of a completely unrelated, much well-characterized viral matrix protein. Whether the same or even a similar image analysis approach (i.e., assigning the outer six-pixel as the PM signal) is applicable to HSPA1A should be the main issue that the authors need to clearly study and thoroughly verify in the current manuscript. Furthermore, in Fig.1,  Fig. 4, Fig. S1 and Fig S2, the implementation of the "+ve PDM" analysis was vaguely described, which raises concern in the interpretation of the data. What did the authors mean by green and red fluorescence? Did imply EGFP and wheat germ agglutinin lectin (WGA) fluorescence? If so, how was WGA signal acquired and analyzed? What exactly is the significance of incorporating the "+ve PDM" analysis?

2) In addition to using EGFP as a negative control for PM localization,  the authors should also employ a positive control experiment for PM localization, such as using the WGA signal that should be the "real" PM signal. The authors should be able to quantitatively determine the CTCF ratio (PM/total) of WGA, which in turn will be useful in assessing the significance in the difference in the CTCF ratios between EGFP and GFP-HSPA1A.

3) Based on the labeling mechanism of WGA, WGA fluorescence is not expected to be notably affected by the presence of HSPA1A. Nevertheless, if one is to closely inspect the representative data shown in both Fig. 1A and Fig. S1A, WGA fluorescence seems  to substantially increase in the presence of HSPA1A at 42°C, but not for EGFP at 42°C. Unless there is a mechanistic link between WGA fluorescence and PM localization of HSPA1A, one cannot help but questioning whether cell samples incubated at 42°C might be associated with certain artifacts that result in non-specific enhancement of WGA fluorescence intensity at the PM, which then raises serious concern about the validity of the authors' observation that GFP-HSPA1A  fluorescence at the PM was increased at 42°C.

4) The authors applied both GFP- and RFP-tagged HSPSA1A for surface biotinylation analyses in Fig. 5 and Fig. S5, respectively. In both cases, the expected molecular weight (MW) of the tagged protein should closed to about 100 kDa. In Fig. 5, indeed, distinct GFP_associated protein bands were noted at about 100 kDa. In Fig. S5 (wherein GFP-HSPA1A was employed), however, the authors showcased two populations of RFP-associated protein bands, the apparent MW of which were both significantly smaller than 75 kDa. No explanation was provided for this serious discrepancy. In fact, if one is to read carefully the Figure legend for Fig S5, the authors even referred to a non-existing "western blots shown in Fig. 3". Overall, it is genuinely difficult not to question the reliability of their surface biotinylation data.

Author Response

Reviewer 4

Yes

Can be improved

Must be improved

Not applicable

Does the introduction   provide sufficient background and include all relevant references?

( )

( )

(x)

( )

Is the research design   appropriate?

( )

( )

(x)

( )

Are the methods   adequately described?

( )

( )

(x)

( )

Are the results   clearly presented?

( )

( )

(x)

( )

Are the conclusions   supported by the results?

( )

( )

(x)

( )

Comments and Suggestions for Authors

This manuscript aims to provide evidence supporting the notion that the heat shock protein HSPA1A may associate with the plasma membrane lipid phosphotidylserine (PS) in a temperature-dependent manner. The majority of the data presented is based on confocal image analyses,  which is complimented with limited supporting biochemical evidence toward the end of the Results section. Although the overall design of the experiments appears to be reasonable, this manuscript lacks sufficient detail and control experiment in support of the rationale for their image analysis protocols and data interpretation. Moreover, for surface biotinylation analyses, a significant  discrepancy was observed that raises serious concern about the quality of their biochemical data. Listed below are some of my major concerns regarding this manuscript.

We would like to thank the reviewer for taking the time to read our manuscript and providing very timely and critical feedback in improving its quality.

1)   Given that virtually all the evidence is based on confocal image analyses, the authors are obligated to provide a comprehensive description of their image acquisition and analysis protocols in the Methods section, which however is lacking. For example, regarding the definition and quantification of the so-called plasma membrane (PM) pixels, readers are expected to  look it up in a previous publication (ref. 29) by one of the co-authors that addresses the membrane lipid-association of a completely unrelated, much well-characterized viral matrix protein. Whether the same or even a similar image analysis approach (i.e., assigning the outer six-pixel as the PM signal) is applicable to HSPA1A should be the main issue that the authors need to clearly study and thoroughly verify in the current manuscript. Furthermore, in Fig.1,  Fig. 4, Fig. S1 and Fig S2, the implementation of the "+ve PDM" analysis was vaguely described, which raises concern in the interpretation of the data. What did the authors mean by green and red fluorescence? Did imply EGFP and wheat germ agglutinin lectin (WGA) fluorescence? If so, how was WGA signal acquired and analyzed? What exactly is the significance of incorporating the "+ve PDM" analysis?

Response: We apologize for these issues and omissions in the original manuscript. In the revised manuscript, we expanded this section in the materials and methods. We further describe the corrected total cell fluorescence (CTCF; described in Burgess et al. 2010-Proc Natl Acad Sci USA 107:12564–12569), which was used to determine the PM localization of HspA1A. Additionally, we clarify that we did not use colocalization or Intensity Correlation Analysis (Li et al. 2004-J Neuroscience 24:4070-4081) to quantify the changes in HspA1A’s PM localization. We also provide further description on the +ve PDM values. We incorporated the PDM images in the figures for two reasons. First, they are clear to see in the black and white color scheme. Second, they provide a quantitative idea of the correlation between the two original images (GFP and RFP) than image superimposition alone.

Based on how we obtained the images the width of the PM (measured using the WGA stain) is approximately 6 pixels. Therefore, we believe that it was reasonable to analyze the 6 outer pixels of each cell. Furthermore, taking into account the biotinylation data, it seems reasonable to suggest that this approach worked in the case of HspA1A.

2) In addition to using EGFP as a negative control for PM localization,  the authors should also employ a positive control experiment for PM localization, such as using the WGA signal that should be the "real" PM signal. The authors should be able to quantitatively determine the CTCF ratio (PM/total) of WGA, which in turn will be useful in assessing the significance in the difference in the CTCF ratios between EGFP and GFP-HSPA1A.

Response: We thank the reviewer for this excellent suggestion. We provide these analyses in the revised Fig.1.

3) Based on the labeling mechanism of WGA, WGA fluorescence is not expected to be notably affected by the presence of HSPA1A. Nevertheless, if one is to closely inspect the representative data shown in both Fig. 1A and Fig. S1A, WGA fluorescence seems  to substantially increase in the presence of HSPA1A at 42°C, but not for EGFP at 42°C. Unless there is a mechanistic link between WGA fluorescence and PM localization of HSPA1A, one cannot help but questioning whether cell samples incubated at 42°C might be associated with certain artifacts that result in non-specific enhancement of WGA fluorescence intensity at the PM, which then raises serious concern about the validity of the authors' observation that GFP-HSPA1A  fluorescence at the PM was increased at 42°C.

Response: We understand this concern. The pattern of the dye used did not depend on the temperature or the construct used. We show these data in the revised Figure 1. However, we feel that it is important to note that we did not use the fluorescence values of the WGA to detect co-localization with A1A. The sole purpose of the WGA staining was to mark the PM. Therefore, we do not believe there is any concern on the CTCF values we present and the conclusions we make. We clarified these issues in the revised manuscript.

4) The authors applied both GFP- and RFP-tagged HSPSA1A for surface biotinylation analyses in Fig. 5 and Fig. S5, respectively. In both cases, the expected molecular weight (MW) of the tagged protein should closed to about 100 kDa. In Fig. 5, indeed, distinct GFP_associated protein bands were noted at about 100 kDa. In Fig. S5 (wherein GFP-HSPA1A was employed), however, the authors showcased two populations of RFP-associated protein bands, the apparent MW of which were both significantly smaller than 75 kDa. No explanation was provided for this serious discrepancy. In fact, if one is to read carefully the Figure legend for Fig S5, the authors even referred to a non-existing "western blots shown in Fig. 3". Overall, it is genuinely difficult not to question the reliability of their surface biotinylation data.

Response: We believe that this is a misunderstanding as a result of a miscommunication from our part in the original submission. We apologize for this miscommunication of the experiments. The experiments were performed using the HspA1A-GFP protein. The blots that were blotted with the anti-RFP correspond to the empty RFP vector and Lact-C2-mCherry proteins and do not represent HspA1A, which in these experiments was tagged with GFP (HspA1A-GFP). Therefore, we do not believe there is a discrepancy. We revised the text and legends as well as the figures to ensure clarity over this issue. We also removed the erroneous sentence from the legend.

Reviewer 5 Report

In the manuscript by Bilog et al., authors demonstrate that HspA1A translocates to the plasma membrane of cells upon/after heat shock treatment. The association of HspA1A with the plasma membrane depends on the PS. The manuscript is well writen and presented data are solid and of sufficient quality.

The weakest part of the manuscript in my opinion is description of HspA1A integration into the plasma membrane studied with surface-biotinylation. This part requires more detailed description of eperimental model and interpretation of obtained results. The important controls for the experiment should be moved from Supplementary materials into Figure 5.

The biotinylation data suggest that HspA1A fully integrates into the PM and even partially translocates through the PM, as it is accessible for modification that takes place in the extracellular space. Can authors discuss what is the molecular mechanism of such phenomenon? Can HspA1A be subsequently released from the PM to the extracellular space?

Author Response

Reviewer 5

  Yes

Can be improved

Must be improved

Not applicable

Does the introduction provide   sufficient background and include all relevant references?

(x)

(   )

(   )

(   )

Is the research design   appropriate?

(x)

(   )

(   )

(   )

Are the methods adequately   described?

(x)

(   )

(   )

(   )

Are the results clearly presented?

(x)

(   )

(   )

(   )

Are the conclusions supported by   the results?

(x)

(   )

(   )

(   )

Comments and Suggestions for Authors

In the manuscript by Bilog et al., authors demonstrate that HspA1A translocates to the plasma membrane of cells upon/after heat shock treatment. The association of HspA1A with the plasma membrane depends on the PS. The manuscript is well writen and presented data are solid and of sufficient quality. 

The weakest part of the manuscript in my opinion is description of HspA1A integration into the plasma membrane studied with surface-biotinylation.

We would like to thank the reviewer for taking the time to read our manuscript and providing very timely and critical feedback in improving its quality.

This part requires more detailed description of eperimental model and interpretation of obtained results.

The important controls for the experiment should be moved from Supplementary materials into Figure 5. 

Response: We thank the reviewer for this comment. In the revised manuscript, we updated the figure as per the reviewer’s request, which now includes the control antibodies. However, because the resulting figure (including all the experiments) was “busy”, we decided to separate the blots based on the presence or absence of sphingosine.

The biotinylation data suggest that HspA1A fully integrates into the PM and even partially translocates through the PM, as it is accessible for modification that takes place in the extracellular space. Can authors discuss what is the molecular mechanism of such phenomenon? Can HspA1A be subsequently released from the PM to the extracellular space?

Response: We appreciate this excellent comment. Unfortunately, we do not know the exact molecular mechanism or whether protein is subsequently released to the extracellular space. Nevertheless, as per the reviewer’s request, we added a short paragraph providing a speculative scenario.

Round 2

Reviewer 1 Report

Unfortunately, the rebuttal does nothing to assuage my initial doubts about the central conclusion of the paper, namely that it is specifically PS that targets HspA1A to the PM. The experiments suggested as far from outside the scope of the manuscript - to be certain it is genuinely a selective PS interaction, the other negatively charged lipids must be ruled out.

Note, my last comment as cut off from the web interface. My question was whether the cells were imaged live in Fluormount G? This is an aqueous/glycerol based medium that will induce osmotic perturbations if the cells are not fixed.

Reviewer 2 Report

None

Reviewer 3 Report

The authors has addressed all the issues

Reviewer 4 Report

The authors have addressed my concerns. I have no further comment.